# To Flee or Not to Flee: How Age, Reproductive Phase, and Mate Presence Affect White Stork Flight Decisions

**DOI:** 10.3390/ani13182920

**Published:** 2023-09-14

**Authors:** Laïd Touati, Mohamed Athamnia, Abdennour Boucheker, Bourhane-Edinne Belabed, Farrah Samraoui, Ahmed H. Alfarhan, Anders P. Møller, Boudjéma Samraoui

**Affiliations:** 1Laboratoire de Recherche et de Conservation des Zones Humides, University of Guelma, Guelma 24000, Algeria; laidbio@hotmail.com (L.T.); athamnia.mouh@yahoo.fr (M.A.); babdennour2007@yahoo.fr (A.B.); fsamraoui@gmail.com (F.S.); 2Biology and Plant Ecology Department, Mentouri Brothers Constantine 1 University, Constantine 25000, Algeria; 3Department of Ecology, University 8 Mai 1945, Guelma 24000, Algeria; 4Department of Biology, University Badji Mokhtar, Annaba 23000, Algeria; bourhanebelabedmarine@yahoo.fr; 5Department of Botany & Microbiology, College of Science, King Saud University, P.O. Box 2455, Riyadh 11451, Saudi Arabia; alfarhan@ksu.edu.sa; 6AgroParisTech, Ecologie Systématique et Evolution, Université Paris-Saclay, CNRS, 91190 Gif-sur-Yvette, France; anders.moller@u-psud.fr

**Keywords:** disturbance, flight initiation distance, parental care, reproductive strategies, sexual conflict, wildlife management

## Abstract

**Simple Summary:**

High levels of predation can reduce survival rates of young birds and affect overall population growth. Therefore, birds that are better adapted to avoid predators, such as through strategic nest positioning, camouflage, and loud alarm calls, have a greater chance of successfully fledging their young. This study examines how white storks adapt to potential human threats during their nesting season and shows the effects of factors such as age, reproductive stage, and presence of a mate on their nesting behavior. The results show that storks are able to adapt their defense strategies depending on the perceived value and level of threat to their current brood. In particular, during crucial breeding phases, storks tend to prolong their stay in the nest while accelerating their return, reflecting a delicate balance between immediate reproductive needs and future prospects. In addition, the influence of a mate leads to earlier departure from the nest, suggesting a possible sexual conflict and interplay between parental care priorities. These results provide a deeper understanding of the intricate decision-making mechanisms of white storks when faced with perceived threats during the breeding season. The study contributes to a more comprehensive understanding of avian behaviors in response to environmental challenges.

**Abstract:**

Recognizing, assessing, and responding to threats is critical for survival in the wild. Birds, especially in their role as parents, must decide whether to flee or delay flight when threatened. This study examines how age, reproductive stage, and the presence of a mate influence flight initiation distance (FID) and nest recess duration in white storks. Analyzing the data with a generalized additive mixed model (GAMM), we found significant correlations between FID and age, reproductive stage, and presence of a mate. These results suggest that the trade-off between current and future reproduction shifts during critical breeding periods, such as incubation and nestling care. To increase breeding success, White Storks appear willing to take risks and extend their stay in the nest when offspring are most valuable and vulnerable. In the presence of a mate, individuals leave the nest earlier, suggesting possible sexual conflict over parental care. The duration of nest abandonment is consistent with FID, except for age. These results illustrate how parental age, brood value, vulnerability, and sexual dynamics influence white stork flight decisions in complex ways. Understanding these dynamics enriches our knowledge of bird behavior and adaptations to environmental challenges and highlights the complexity of parental decision making.

## 1. Introduction

Breeding success depends on several factors, including the birds’ ability to find suitable nesting sites, secure sufficient food for themselves and their chicks, and protect their offspring from predators. Birds that are better adapted to avoid predators, such as through effective nest placement, camouflage, or alarm calls and mobbing, are more likely to fledge their offspring. High predation rates can reduce the number of chicks that survive to adulthood, thus affecting overall population growth. As a result, predation mortality is an important selective force and a critical determinant of breeding success in birds because it influences bird behavior, morphology, and reproductive strategies [1,2]. Therefore, offspring defense against predation is an important component of parental investment in many species, with potential fitness costs [3,4,5]. In addition, the intensity of nest defense has been shown to be positively related to breeding success [6,7].

Theory predicts that in altricial birds, the reproductive value of offspring monotonically increases with age and parents invest more in the current brood when the prospect of nesting again during the season decreases [8,9]. Conversely, theory also predicts that parents of precocial birds invest less in defense after their young hatch and disperse [10]. However, there are inter- and intraspecific differences in parental defense, and many factors may influence the residual reproductive value. Offspring value or parental condition may decline over the course of the breeding season [11,12,13]. Other factors such as parental uncertainty may cause males to defend less [14].

When animals accurately identify potential threats, they can take appropriate action to avoid dangerous situations, which increases their chances of survival. Flight initiation distance (FID) is defined as the distance an individual will tolerate before taking flight from an approaching threat, and it is a useful measure of risk-taking behavior [15]. The decision-making process for flight initiation distance has been theoretically evaluated but not yet adequately tested [16,17]. 

The decision to flee or suppress the escape response may depend on a number of factors that determine the benefits and costs of antipredator behavior [18]. In addition to natural predators, animals must contend with an ever-increasing human population [19,20]. However, due to the complexity of the escape response, which includes subtle and nonlethal effects [21,22], assessing the effects of human disturbance on animals remains challenging [23].

FID and duration of absence are behaviors that can occur in a variety of situations (e.g., foraging) and they can be considered to be defensive of the nest and offspring in the context of nesting [15]. In addition, numerous studies have shown that predation and FID are closely linked, with prey often adapting their response to predator behavior [24]. The effects of predation and disturbance on animal behavior and population dynamics can have far-reaching consequences beyond changes in FID. For example, in avian species, hunting pressure can affect nest presence and ultimately influence breeding success [25]. A higher nest presence positively affects breeding success in several bird species [26,27]. Conversely, nest recess has been associated with breeding failure [28,29,30], suggesting that nest absence is associated with fitness costs [31].

The persistent and cumulative effects of human disturbance have been shown to have a range of negative impacts on animals that ultimately negatively affect their fitness and long-term viability [32,33]. These studies show that the negative effects of human disturbance go beyond physical damage or habitat destruction and can include behavioral changes, physiological stress, and reduced reproductive success [34,35,36]. 

Various hypotheses, such as offspring value, vulnerability, parental condition, etc., provide a conceptual framework to understand the complex dynamics of parental investment in the context of sexual selection [37,38]. Some of these hypotheses used to explain nest defense are not mutually exclusive, and their relative importance may vary depending on the specific ecological conditions, evolutionary history, and behavioral strategies of different species [39].

In the Mediterranean, white storks (*Ciconia ciconia*) often breed in olive groves or rural areas and are, therefore, vulnerable to aerial predators as well as heat stress and bad weather. They are also exposed to anthropogenic disturbance. A previous study examining white storks’ responses to drones found that these devices are not perceived as particularly threatening [40].

Theory suggests that white storks are more sensitive to perceived threats at certain stages of the breeding cycle and may exhibit longer FIDs and reduce the duration of nest recess during egg-laying, incubation, and chick brooding. In addition, age and the presence of a mate may influence these responses. The objectives of our study were, first, to determine FID and the duration of nest recess of nesting white storks disturbed by an approaching human, and, second, to examine whether differences in parental decisions related to age, timing of breeding, and the number of adults occupying the nest could be interpreted within the framework of life history theory [3].

## 2. Materials and Methods

### 2.1. Study Area

Algeria hosts a substantial population of white storks that have formed colonies on the southern edge of their breeding range, dotting the landscape with loose settlements. One of these colonies, a thriving one, has settled in an olive grove near Dréan in northeastern Algeria (36° 41.1700′ N, 7° 41.5200′ E). Interestingly, this colony is located only 300 meters from the largest garbage dump in the region. The breeding season at this Dréan site extends from late February to mid-July. Since 2011, the breeding ecology of this white stork population has been meticulously monitored. Each year, chicks are fitted with uniquely coded Darvic PVC rings prior to fledging as part of a chick ringing program. Our experiment was conducted at this colony (Figure 1a), which hosts approximately two hundred breeding pairs annually and has served as a study site for ecological studies since 2011 [41,42].

White storks often show a preference for rural habitats, nesting near human settlements and using man-made structures such as roofs and poles. In Dréan, these birds prefer to nest on olive trees (*Olea europaea* L.). The olive grove also serves as pasture, mainly for sheep and cows, and is surrounded by cultivated fields. White stork nests are subject to potential threats from a variety of sources, including roaming children, common ravens (*Corvus corax*), black kites (*Milvus migrans*), and booted eagles (*Hieraaetus pennatus*).

The Dréan olive grove, located close to human activities, provides a unique opportunity to explore the intricate interactions and adaptations of white storks in an environment they share with both their human neighbors and potential predators. This site is an important arena for understanding the complex dynamics that govern stork behavior and reproductive success amidst various challenges.

### 2.2. Data Collection

Trees were evenly distributed in a 10 m grid and only those that contained a single nest (which constituted the majority) were selected. Specifically, only trees with a height of 5 to 6 m were retained. Nests were sampled between 29 February and 29 June 2018. We estimated FIDs for nesting white storks in a standardized manner: a researcher, always the same, moved at normal speed with his head toward an occupied nest and recorded with a surveyor’s (measuring) rope the distance at which the bird initiated flight distance. The experiment focused on ringed individuals. The starting distance was not recorded, but we ensured that it was over 100 m. The duration between the bird’s departure and return was also noted. If a bird did not return after half an hour, it was recorded as absent for 30 min. 

### 2.3. Statistical Analysis

We ran a generalized additive mixed model (GAMM) with a Gaussian error distribution to test whether the flight initiation distance (response variable) varied as a function of the date (1 = 1 January), age, number of adults in the nest, time of day, and breeding success (explanatory variables). Similarly, we ran a GAMM with a Gaussian error distribution to test whether the duration of nest recess was also related to the same covariates. We used the bird ID (ringed bird) as a random effect. We also tested whether the time span of nest recess was related to FID by running a generalized linear model (GLM) with a binomial error distribution and a logit link function. The time delay was categorized as “Late” if the delay was over 30 min and “Early” if the delay was under 30 min. 

To test whether the response variables (FID and leave duration) could be explained by reproductive phase, we performed another GAMM by dividing the sampling date into four periods: pre-egg-laying, egg-laying and incubation (32 days), chick brooding (28 days), and post-brooding. We retained the other two explanatory variables of age and presence of a mate. All statistical analyses were performed using R software (version 4.2.3) [43].

## 3. Results

We collected 287 samples from 59 different white storks ranging in age from 2 to 7 years. Age is an important determinant of arrival time in white storks, with younger birds arriving and breeding later in the colony than older birds, resulting in a longer breeding season (Figure 1b). FID ranged from 0.0 m (birds that remained in the nest) to 57.8 m, with a mean of 17.4 ± 11.9 m. Time spent away from the nest ranged from 0.0 min to 30.0 min (birds that did not return to the nest after half an hour), with a mean of 7.5 ± 9.6 min.

The pattern of FID was described as a U-shape, with a decrease in distance from the beginning of the study to mid-March. FID stabilized at this minimum value until early May and then gradually increased until the end of the study (Figure 2a). FID was also significantly correlated with age, with younger birds initiating a flight response at a greater distance from the experimenter (Table 1). In addition, birds flew at a greater distance when their mate was in the nest (Table 1).

The duration of nest recess also followed a U-shape, with a gradual decrease from the beginning of the study to the minimum between mid-March and early May before increasing again until the end of the experiment in late June (Figure 2b). Age was negatively correlated with the duration of nest recess, but not in a significant way. In contrast, the presence of a mate significantly increased the duration of nest recess (Table 1).

In addition, the duration of nest recess was significantly correlated with FID (Table 2). The GLM results showed that FID was around 25 m; birds with a larger FID took increasingly longer to return to their nests (Figure 3).

To confirm our preliminary results that FID might be closely related to the reproductive phase of the birds studied, we tested this hypothesis by replacing the sampling date (time of the year) with the four reproductive phases (Figure 4) and used a GAMM to observe if the response variables (FID and nest recess) could be explained by age, presence of a mate, and reproductive phase. The results showed that age, presence of a mate, and reproductive phase all played a significant role in explaining FID (Table 3). Similarly, nest recess could also be explained by the presence of a mate and reproductive phase. Egg-laying and incubation as well as chick brooding were the two periods with the shortest nest recess (Table 3).

## 4. Discussion

The results show that adult white storks are able to adapt their behavior to perceived threats. Being present in the nest allows the birds to protect their clutch and brood, and meet the thermal needs of their eggs and nestlings. When confronted with drones instead of humans, adult birds had a shorter FID (20 m). They also returned earlier (23 s on average) after disturbance by drones [40]. Such adaptations of nest defense behavior are consistent with the risk to parents’ hypothesis, which posits that parents adapt their behavior to balance the benefits of the reproductive value of the current offspring against the costs to their own survival and future reproductive opportunities [44].

The results also show a significant correlation between FIDs of breeding white storks and the date (1 = 1 January) of the year, age, and presence or absence of a breeding partner. Egg-laying occurs between the second half of February and early May, and breeding date is strongly associated with arrival date and age [41,42,45]. The significant relationship between FID and reproductive stage found in white storks ([40], this study) and great egrets (*Ardea alba*) [46] is partially consistent with the offspring value hypothesis [10,47], which states that parents make decisions that maximize their overall reproductive fitness, including both current and future reproductive success. However, during the late rearing phase, which is characterized by frequent agonistic behavior of white stork fledglings [40], the increase in FID is not consistent with the above hypothesis.

In contrast, the results support the vulnerability hypothesis [48], according to which birds have a shorter FID when incubating or brooding nestlings. They also leave the nest quickly before egg-laying or when they have older chicks, suggesting that white storks protect their nest during critical periods (incubation and brooding young chicks) by staying in the nest longer. Thus, parents optimally adapt their nest defense to the developmental stage of their offspring [40]. As predicted, age has a significant influence on when birds decide to leave their nest. Young individuals tend to have a longer FID, indicating a higher residual reproductive value, while older birds prefer to wait longer before leaving the nest. Younger stork pairs breed later in the season than older pairs and often have lower nesting success [41]. Thus, younger pairs may not place as much value on their broods as older pairs [49]. Nest defense also increases with parental age in California gulls (*Larus californicus*) [50]. However, parental age is often closely related to experience, reproductive characteristics, and survival, making it difficult to tease apart the various costs and benefits of parental decisions [39,51,52].

The presence of a partner also has a significant effect on the distance at which white stork parents leave their nests. In the presence of a partner, individuals reduce their risk-taking by choosing a longer FID and leaving their mate to bear the cost of protecting the nest. This result is consistent with the theory that the sexes have conflicting evolutionary interests [53].

In addition, this study shows a strong correlation between the amount of time individuals leave the nest unattended after human disturbance and FID. White storks with shorter FIDs are more likely to return to their nests after shorter periods of time when disturbed, highlighting the need to reduce threats to nests during critical periods [48]. As noted for FID, the reproductive phase—including incubation and nestling care—tends to shorten the time away from the nest. This behavior may be due to the protective instinct of individuals that are reluctant to leave the nest during periods of high offspring value [10] and vulnerability [48]. These results suggest that nest defense is positively correlated with brood vulnerability and decreases as chicks grow and threats diminish [40,54,55].

Age does not correlate with the duration of nest absence, unlike the presence of a mate, which tends to increase the duration of nest recess. As parental care is costly, an individual’s decision to leave the nest earlier in the presence of its mate suggests sexual conflict over parental care [56,57].

Predator defense responses such as flight in nesting white storks can be viewed as a cost-benefit trade-off between reduced predation risk and increased vulnerability of the clutch and brood [17,48]. This response not only diverts energy expenditure from brooding or resting, but also exposes eggs and chicks to adverse weather conditions and predators. Heat stress or cold snaps can impair egg development or kill young chicks that are unable to thermoregulate [58,59]. These results are also consistent with several studies that have shown that human disturbance has the greatest negative impact on breeding success during the egg and chick stages [60,61,62].

The spread of human settlements and the conversion of natural habitats to urban landscapes have impacted white stork survival strategies through influences such as invasive predators and urbanization. Human presence has altered the selective pressures that drive white stork behavior. With the increasing prevalence of invasive predators, white storks must navigate a changing landscape of threats—a landscape that requires adaptations to both traditional and invasive predators [63]. In rural areas, habituation or other selection pressures have led successful white storks dwellers to modify their behavior toward predators [64] and adapt (reduce) flight initiation distance, possibly at a cost [65,66]. Therefore, our study needs to be replicated under different habitat conditions to examine the range of behavioral plasticity of white storks.

A comprehensive understanding of the behavioral plasticity of defense mechanisms is of great importance, especially for species threatened with extinction. This knowledge becomes a compass to guide conservation efforts and provide insights into the survival strategies of endangered species. In a world where ecotourism has become increasingly important [67], protected areas where these species are conserved are under increasing pressure from human activities [68]. The delicate balance between conservation of these habitats and increasing human presence requires innovative solutions. In this complex interplay between human disturbance and ecological integrity, the use of FID is emerging as an important tool. Wildlife managers charged with protecting wildlife have used FID as a tool to design and implement buffer zones [69,70,71]. FID is becoming the critical metric that delineates these vital spaces and creates an area where human and wildlife needs can coexist.

## 5. Conclusions

In summary, this study reveals the impressive adaptability of adult white storks in the face of perceived threats and sheds light on the intricate mechanisms underlying their survival strategies. These storks, whose primary focus is protecting their nests and offspring, show nuanced adaptations to various disturbances, highlighting the complexity of their interactions in their environment. In particular, comparison of disturbance by approaching humans and flying drones reveals specific behavioral responses towards humans: an intentional extension of flight distance (FID) followed by a delayed return to the nest. This complicated interplay of behaviors is consistent with the risk to parents’ hypothesis and highlights the intricate balance between parental survival and offspring reproductive potential.

The study establishes a clear relationship between FID, reproductive stages, age, and the presence of a mate. Longer nest presence is consistent with the principles of the vulnerability hypothesis. This results in shorter FIDs during incubation and nestling care and underscores the adaptive focus on protecting the most vulnerable life cycle stages. The nuanced variations at FID, corresponding to different breeding phases, provide insight into residual reproductive value and illustrate the plasticity of decision making in these storks. However, the presence of a mate introduces an additional layer of complexity—a mixture of mutual benefit and conflict. This complexity in adult decision making, which is strongly influenced by age, experience, and reproductive stage, is enriched by these nuanced relationships.

The implications of this study extend far beyond white stork behavior and reproductive strategy to shed broader light on the field of conservation. In an era characterized by burgeoning ecotourism and increasing human pressures, the protection of wildlife habitats assumes paramount importance. The management of human-induced disturbance is becoming increasingly important and requires the formulation of strategies that balance human activities with wildlife welfare. The effectiveness of FID as a tool for establishing buffer zones to promote wildlife conservation is becoming increasingly evident. These findings underscore the ever-evolving synergy between scientific knowledge and practical conservation that helps maintain fragile coexistence with the natural world.

## Figures and Tables

**Figure 1 animals-13-02920-f001:**
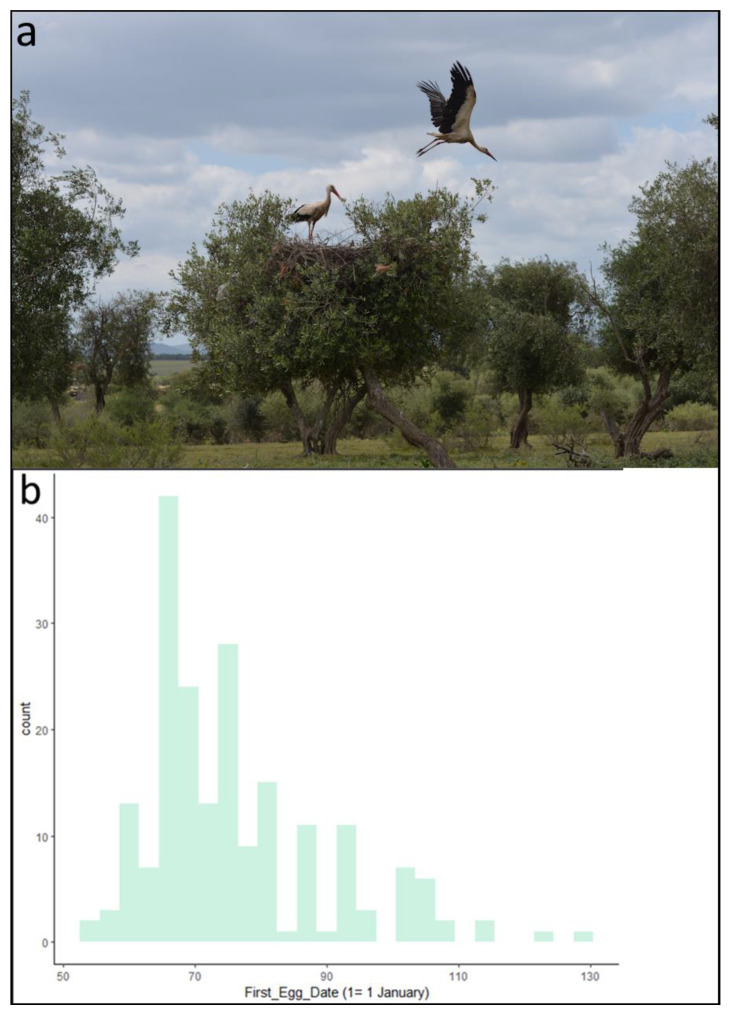
(**a**) View of the white stork colony at Dréan, northeastern Algeria; (**b**) histogram of egg-laying dates by white storks at Dréan.

**Figure 2 animals-13-02920-f002:**
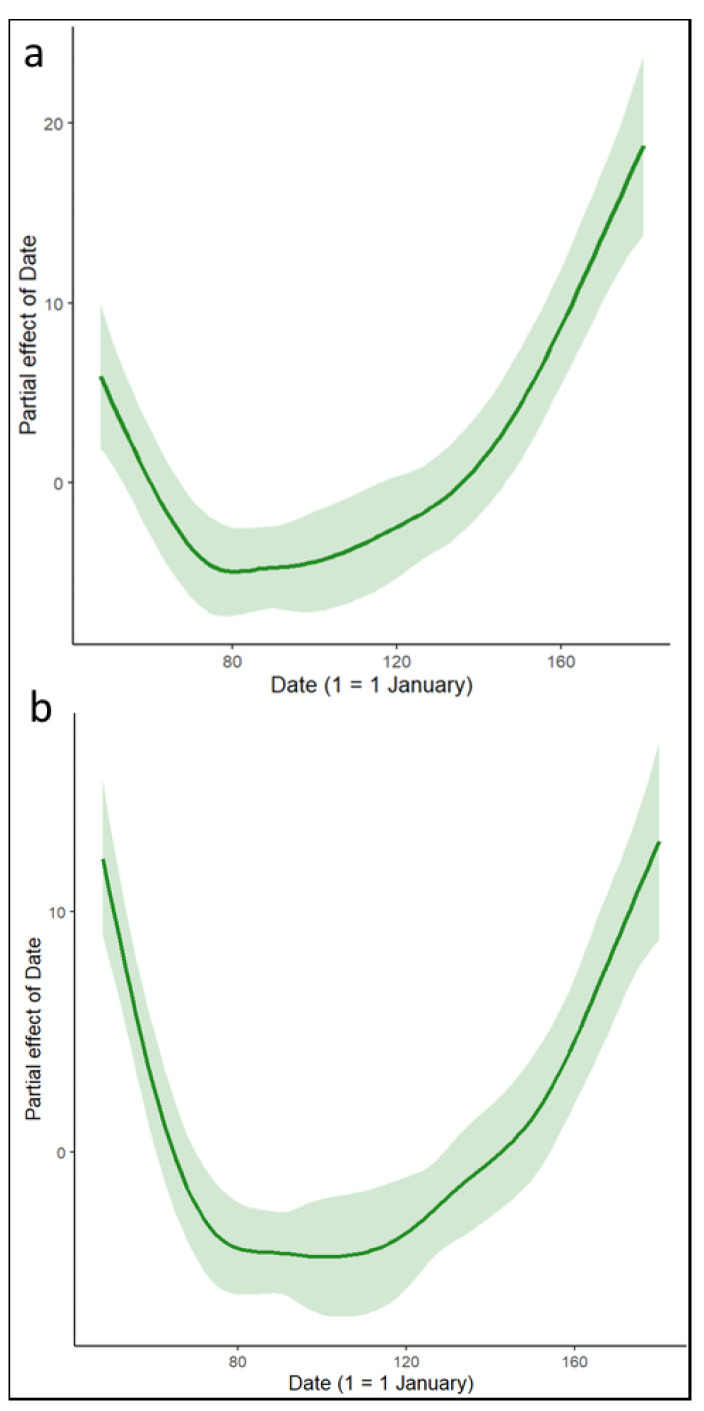
Fitted curves for the Gaussian GAMM displaying the relationship between (**a**) flight initiation distance and date (1 = 1 January) and (**b**) duration of nest recess and date (1 = 1 January). Shaded areas represent 95% confidence intervals.

**Figure 3 animals-13-02920-f003:**
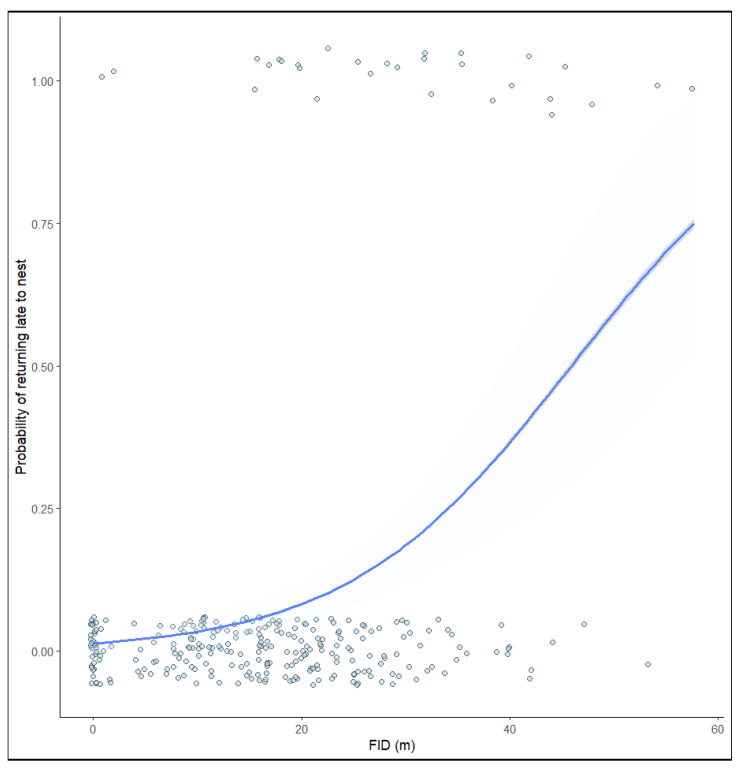
Fitted curve depicting the relationship between the probability of returning late to the nest and FID. Shaded areas represent 95% confidence intervals.

**Figure 4 animals-13-02920-f004:**
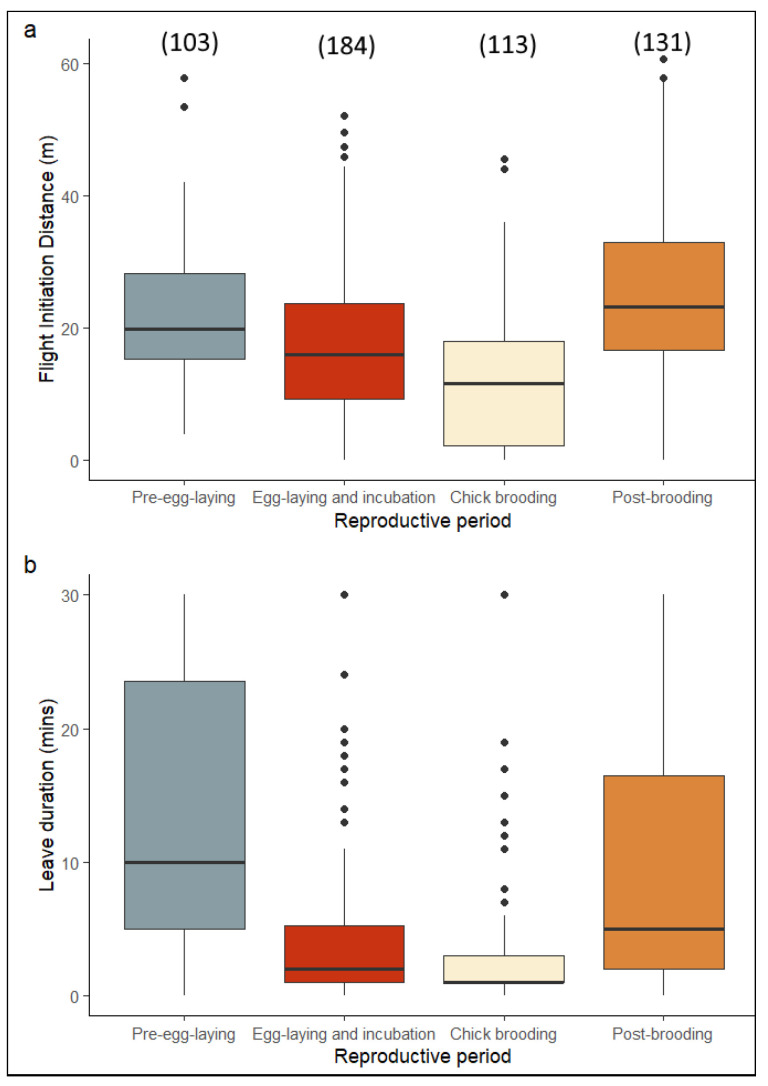
Boxplots of flight initiation distance (**a**) and duration of nest recess (**b**) during the four reproductive phases of white storks. Values are medians and 95% CIs. The sample size is given in parentheses.

**Table 1 animals-13-02920-t001:** Parameter estimates for GAMMs of flight initiation distance and period of nest recess of disturbed white storks. The terms edf and Ref.df stand for effective degree of freedom and reference degree of freedom, respectively.

Parameter	Estimate	SE	*t*-Value	*p*-Value
	Flight Initiation Distance			
Parametric Coefficients				
Intercept	12.26	3.4	3.61	0.0004
Age	−1.36	0.6	−2.26	0.0245
No. of Individuals	8.27	1.3	6.36	8.10 × 10^−10^
	edf	Ref.df	Chi.sq	*p*-Value
Smooth Terms				
Date	4.64	4.64	24.56	<2× 10^−16^
	Leave Duration			
Parametric Coefficients				
Intercept	4.88	2.32	2.11	0.0362
Age	−0.53	0.37	−1.43	0.1527
No. of Individuals	3.62	1.05	3.45	0.0007
	edf	Ref.df	Chi.sq	*p*-Value
Smooth Terms				
Date	5.49	5.49	26.47	<2× 10^−16^

**Table 2 animals-13-02920-t002:** GLM parameter estimates for the period of nest recess of disturbed white storks as predicted by FID (null deviance = 187.9 anddf = 286; residual deviance = 155.7 anddf = 286).

Predictor	Odds Ratio	95% CI Lower Bound	95% CI Upper Bound	Std. Error	z-Value	Pr
Intercept	0.014	0.005	0.035	0.518	−8.217	<2 × 10^−16^
FID	1.097	1.05	1.148	0.018	5.191	2.09 × 10^−7^

**Table 3 animals-13-02920-t003:** Parameter estimates for GAMMs of flight initiation distance (FID) and nest recess of disturbed white storks during four reproductive phases.

Parameter	Estimate	Se	*t*-Value	*p*-Value
	FID			
Parametric Coefficients				
Intercept	29.27	2.99	9.78	<2 × 10^−16^
Egg-Laying and Incubation	−5.28	1.33	−3.98	7.80 × 10^−5^
Chick Brooding	−7.35	1.63	−4.51	8.20 × 10^−6^
Post-Brooding	7.2	1.62	4.45	1.06 × 10^−5^
Age	−2.31	0.55	−4.18	3.45 × 10^−5^
No. of Individuals	5.46	1.18	4.61	5.10 × 10^−6^
	Nest Recess			
Intercept	11.41	1.78	6.39	3.65 × 10^−10^
Egg-Laying and Incubation	−7.83	1.09	−7.15	2.97 × 10^−12^
Chick Brooding	−7.84	1.34	−5.86	8.18 × 10^−9^
Post-Brooding	−1.26	1.32	−0.96	0.339
Age	−0.18	0.28	−0.63	0.532
No. of Individuals	3.82	0.94	4.05	5.97 × 10^−5^

## Data Availability

Data are available from the authors upon request.

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
