# Peer review of "To Flee or Not to Flee: How Age, Reproductive Phase, and Mate Presence Affect White Stork Flight Decisions"

_animals, 2023, doi:10.3390/ani13182920_

Round 1
Reviewer 1 Report
The authors assessed flight initiation distance (FID) in response to approaching humans in white storks at their nest during the four stages (pre-laying, laying, incubation, brooding) of their breeding period. The results suggests that white storks adjusts their FID in relation to age, breeding stage and presence of a partner in the nest.
The study uses a well-known experimental paradigm to test for FID in breeding birds. The results support the conclusions of a large body of previous work and are overall well-presented and discussed. I have some comments in relation to unclear text in the methods and some confusion with the terminology.
Lines 82-83: of course nest abandonment is associated with breeding failure, so not exactly clear what you mean here. Please rephrase.
Lines 122: what was the nest density at your field site? And the average distance between each sampled nest? From the photo looks like nests can be very close to each other, meaning that testing FID on one nest might also affect nearby nests.
Lines 127-128: I do not understand why the starting distance was not recorded. This could have been achieved very simply by using a laser distance meter, for example the type commonly used by golfers. If this was not measured, how did you “make sure it was over 100m”?
Line 130: not sure what you mean with collecting data “blindly”. Relative to what?
Line 135: how did you quantify the “number of birds in the nest”? Is this adults+chicks? Or only adult individuals, therefore either one individual or a pair in the nest? This is not clear and is confusing also later on in the results.
Line 136: I am assuming that with covariates you are referring to time of day and breeding success. However, please make it explicit in the text otherwise looks like the covariates in brackets refers to all variables in the model.
Line 137: based on the text here looks like you are considering all variables in the model as covariate. However, according to your hypotheses date, age and number birds in the nest are your explanatory variables, whereas time of day and breeding success are your covariates. Please, clarify.
Line 138: rather than “ring” it would be more correct to name this random effect as “bird ID”
Line 152: why is the number of samples 4 times greater than the number of nests? Was each nest sampled multiple times? Or is this because of data from multiple birds in the same nest? Please clarify your sampling technique in the methods.
Line 161: not clear, I think you mean that greater experimenter distances elicited an earlier flight response in young birds
Line 162: there is no mention of “flight distance” in the methods, so how exactly did you measure this? Or you mean that the duration of leave was longer when the partner was in the nest? But then, did the partners always flew away at the same time, or rather one flew and the other stayed? Please, clarify
Line 168-171: therefore, age was NOT correlated with duration of nest absence. Also, here and throughout “nest abandonment” is the wrong terminology as this refers to individuals leaving the nest and abandoning the breeding attempt because of a disturbance. Here you are only referring to temporary absence of the disturbed individual, therefore “nest absence” would be more appropriate.
Line 175: correct the text in the figure legend for b “duration of leaving the date”
Line 178-179: please correct this sentence
Lines 183-186: this belongs to the methods
Lines 188-190: “laying and incubation, as well as chick brooding, were the two periods with the shortest…”. Then, thee periods or two?
Figure 4: what are the numbers in brackets?
Line 206: I think it would be more correct to say “adult birds HAD a shorter FID (20m)”
Line 208: which “nest defence behaviour” are you referring to? Here you are describing storks fleeing the nest and then returning once the threat has gone, so this is not a nest defence behaviour such as for example mobbing a predator or sitting on the eggs when a predator approaches. Please clarify
Line 218: you do not explicitly describe anywhere in the text what the “offspring value hypothesis” is, so it is difficult for the reader to understand why the stork behaviour fits with this only “partially”. Please, clarify this as you already did with the “risk to parents” hypothesis in line 208
Line 223: I feel there is some confusion with the terminology here. The birds would tolerate a shorter distance from a potential threat and therefore have shorted FID, but they do not show tolerance to FID itself. FID is just the distance at which an individual would perceive a potential threat as dangerous and therefore flee away from it. Please, correct.
Line 255: change “better” with “protect their nest for longer”
The overall English is fine, some minor corrections are needed
Author Response
Dear Sir/Madam,
Thank you very much for your comments and helpful suggestions. We have attempted to implement all of your suggestions, and hope that this new version will meet with your approval.
Best regards,
The authors
Reviewer # 1
The authors assessed flight initiation distance (FID) in response to approaching humans in white storks at their nest during the four stages (pre-laying, laying, incubation, brooding) of their breeding period. The results suggests that white storks adjusts their FID in relation to age, breeding stage and presence of a partner in the nest.
The study uses a well-known experimental paradigm to test for FID in breeding birds. The results support the conclusions of a large body of previous work and are overall well-presented and discussed. I have some comments in relation to unclear text in the methods and some confusion with the terminology.
Lines 82-83: of course nest abandonment is associated with breeding failure, so not exactly clear what you mean here. Please rephrase.
Answer: "Nest abandonment" was replaced by "nest recess" .
Lines 122: what was the nest density at your field site? And the average distance between each sampled nest? From the photo looks like nests can be very close to each other, meaning that testing FID on one nest might also affect nearby nests.
Answer: We only focused on one nest at a time and recorded the FID for the target nest only. In addition, only nests with ringed adults were sampled so one can safely rule out that testing FID on one nest affected a nearby nest.
Lines 127-128: I do not understand why the starting distance was not recorded. This could have been achieved very simply by using a laser distance meter, for example the type commonly used by golfers. If this was not measured, how did you “make sure it was over 100m”?
Answer: We did not have a laser distance meter (there are no golf courses in the country) and had to record distances with a rope.
Line 130: not sure what you mean with collecting data “blindly”. Relative to what?
Answer: This sentence has been removed.
Line 135: how did you quantify the “number of birds in the nest”? Is this adults+chicks? Or only adult individuals, therefore either one individual or a pair in the nest? This is not clear and is confusing also later on in the results.
Answer: We meant adults. We have changed "birds" to "adults " in the text.
Line 136: I am assuming that with covariates you are referring to time of day and breeding success. However, please make it explicit in the text otherwise looks like the covariates in brackets refers to all variables in the model.
Answer: Covariates in brackets refer to all variables in the modeL
Line 137: based on the text here looks like you are considering all variables in the model as covariate. However, according to your hypotheses date, age and number birds in the nest are your explanatory variables, whereas time of day and breeding success are your covariates. Please, clarify.
Answer: The term covariates have different meanings and we apologize for the confusion. Here covariates are explanatory variables. We have made changes to the text to clarify this issue.
Line 138: rather than “ring” it would be more correct to name this random effect as “bird ID”
Answer: This suggestion has been implemented.
Line 152: why is the number of samples 4 times greater than the number of nests? Was each nest sampled multiple times? Or is this because of data from multiple birds in the same nest? Please clarify your sampling technique in the methods.
Answer: Yes, nests were sampled at different breeding phases and thus multiple times. This was the reason we used random effects (bird ID).
Line 161: not clear, I think you mean that greater experimenter distances elicited an earlier flight response in young birds
Answer: Yes, we have changed the text to make it clearer.
Line 162: there is no mention of “flight distance” in the methods, so how exactly did you measure this? Or you mean that the duration of leave was longer when the partner was in the nest? But then, did the partners always flew away at the same time, or rather one flew and the other stayed? Please, clarify
Answer: Flight distance is FID (distance from the experimenter at which the bird flew). We have modified the text to make it clearer.
Line 168-171: therefore, age was NOT correlated with duration of nest absence. Also, here and throughout “nest abandonment” is the wrong terminology as this refers to individuals leaving the nest and abandoning the breeding attempt because of a disturbance. Here you are only referring to temporary absence of the disturbed individual, therefore “nest absence” would be more appropriate.
Answer: No, there was no correlation between age and duration of nest recess. We have changed "nest abandonment" to "nest recess ». We have changed the text to clarify it.
Line 175: correct the text in the figure legend for b “duration of leaving the date”
Answer: We have corrected this legend.
Line 178-179: please correct this sentence
Answer: Sorry, but it is not clear what should be corrected in the sentence.
Lines 183-186: this belongs to the methods
Answer: This suggestion has been implemented.
Lines 188-190: “laying and incubation, as well as chick brooding, were the two periods with the shortest…”. Then, thee periods or two?
Answer: Two periods : Laying and incubation were lumped together.
Figure 4: what are the numbers in brackets?
Answer: Sample size is provided in brackets. We added this information in the legend.
Line 206: I think it would be more correct to say “adult birds HAD a shorter FID (20m)”
Answer: This suggestion has been implemented.
Line 208: which “nest defence behaviour” are you referring to? Here you are describing storks fleeing the nest and then returning once the threat has gone, so this is not a nest defence behaviour such as for example mobbing a predator or sitting on the eggs when a predator approaches. Please clarify
Answer: Staying in the nest in the face of perceived threats is a form of passive defence. Some birds do not leave the nest, but stay put. Some birds with nestlings emit alarm calls.
Line 218: you do not explicitly describe anywhere in the text what the “offspring value hypothesis” is, so it is difficult for the reader to understand why the stork behaviour fits with this only “partially”. Please, clarify this as you already did with the “risk to parents” hypothesis in line 208
Answer: This suggestion has been implemented.
Line 223: I feel there is some confusion with the terminology here. The birds would tolerate a shorter distance from a potential threat and therefore have shorted FID, but they do not show tolerance to FID itself. FID is just the distance at which an individual would perceive a potential threat as dangerous and therefore flee away from it. Please, correct.
Answer: The text has been corrected.
Line 255: change “better” with “protect their nest for longer”
Answer: This suggestion has been implemented.

Reviewer 2 Report
Dear Authors
The paper is quite well writen, even it I have some suggestions, as well as things that need to be improve - like citations way or lack inside Methodology chapter.
Because you cooperate in 8 Authors, some of you work at the same University, I suggest - in 12-15 lines to make better order, maybe each ORCID and e-mail in separate line?
Lines 45-59 - I suggest to enrich this part also in new articles, because you cited mainly articles from XX century (except two of them).
From 45 line - whole article - you don't cite papers in way that is required by MDPI. E.g. [1,2], [4-6], etc. It must be changed.
Lines 119-122 - each English name of the species should be written in small letters
Lines 121-131 - "Data collection" - in my opinion this chapter is not written with all needed details. I guess you used drones right? It should be explained with all details. If not, describe your methods with details, because now it is not clear.
Figure 1. Photo - actual quality of the photo is totally bad, please use photo in better quality.
Line 152 - what you mean as "samples"?
Line 153 - white stork, not White Stork. Change it in whole text.
Line 206 and 206 - if you cite Zbytyt from 2021 line by line, it is not nessesary to do it twice, one is enough.
Author Response
Dear Sir/Madam,
Thank you very much for your comments and helpful suggestions. We have attempted to implement all of your suggestions, and hope that this new version will meet with your approval.
Best regards,
The authors
Reviewer # 2
Dear Authors
The paper is quite well writen, even it I have some suggestions, as well as things that need to be improve - like citations way or lack inside Methodology chapter.
Because you cooperate in 8 Authors, some of you work at the same University, I suggest - in 12-15 lines to make better order, maybe each ORCID and e-mail in separate line?
Answer: We apologize, but we have to follow the guidelines provided by the template.
Lines 45-59 - I suggest to enrich this part also in new articles, because you cited mainly articles from XX century (except two of them).
Answer: We have added three references from the XXI century.
From 45 line - whole article - you don't cite papers in way that is required by MDPI. E.g. [1,2], [4-6], etc. It must be changed.
Answer: This suggestion has been implemented.
Lines 119-122 - each English name of the species should be written in small letters
Answer: This suggestion has been implemented.
Lines 121-131 - "Data collection" - in my opinion this chapter is not written with all needed details. I guess you used drones right? It should be explained with all details. If not, describe your methods with details, because now it is not clear.
Answer: No drones were used. We have modified the text to make it clearer.
Figure 1. Photo - actual quality of the photo is totally bad, please use photo in better quality.
Answer: We have changed the photo.
Line 152 - what you mean as "samples"?
Answer: Each experiment (test of a nest) was a sample.
Line 153 - white stork, not White Stork. Change it in whole text.
Answer: This suggestion has been implemented.
Line 206 and 206 - if you cite Zbytyt from 2021 line by line, it is not nessesary to do it twice, one is enough.
Answer: This suggestion has been implemented.

Reviewer 3 Report
Your paper is well written and clearly presented, and I can offer only these few suggestions:
R 55: change “:” to “.”
R 126: You should clarify whether FID distance was recorded using range finders, or just estimated visually
Author Response
Dear Sir/Madam,
Thank you very much for your kind comments and efforts aimed at improving our manuscript. We have implemented all of your suggestions.
With renewed thanks and best wishes,
The authors
